# Supervising the Transfer of Reasoning Patterns in VQA

Corentin Kervadec*[1,2]    Christian Wolf*[2]    Grigory Antipov[1]    Moez Baccouche[1]
Madiha Nadri[3]

[1]Orange Innovation, France    [2]LIRIS, INSA-Lyon, France    [3]LAGEPP, Université de Lyon, France
firstname.lastname@orange.com, christian.wolf@insa-lyon.fr,
madiha.nadri@lagep.univ-lyon1.fr,  *equal contribution

## Abstract

Methods for Visual Question Anwering (VQA) are notorious for leveraging dataset biases rather than performing reasoning, hindering generalization. It has been recently shown that better reasoning patterns emerge in attention layers of a state-of-the-art VQA model when they are trained on perfect (oracle) visual inputs. This provides evidence that deep neural networks can learn to reason when training conditions are favorable enough. However, transferring this learned knowledge to deployable models is a challenge, as much of it is lost during the transfer. We propose a method for knowledge transfer based on a regularization term in our loss function, supervising the sequence of required reasoning operations. We provide a theoretical analysis based on PAC-learning, showing that such program prediction can lead to decreased sample complexity under mild hypotheses. We also demonstrate the effectiveness of this approach experimentally on the GQA dataset and show its complementarity to BERT-like self-supervised pre-training.

## 1 Introduction

Reasoning over images is the main goal of Visual Question Anwering (VQA), a task where a model is asked to answer questions over images. This problem is a test bed for the creation of agents capable of high-level reasoning, as it involves multi-modal and high-dimensional data as well as complex decision functions requiring latent representations and multiple hops. State-of-the-art models are notorious for leveraging dataset biases and shortcuts in learning rather than performing reasoning, leading to lack of generalization, as evidenced by extensive recent work on bias oriented benchmarks for vision-and-language reasoning [1, 20, 21, 31]. Even large-scale semi-supervised pre-training methods, which successfully managed to increase overall VQA performance, still struggle to address questions whose answers are rare given a context [20].

It has been recently shown that reasoning patterns emerge in attention layers of a SOTA VQA model when trained on perfect (oracle) visual inputs, which provides evidence that deep neural networks can learn to reason, when training conditions are favorable enough [21]. In particular, uncertainty and noise in visual inputs seems to be a major cause for shortcut learning in VQA. While this kind of methods provide strong empirical results and insights on the bottlenecks in problems involving learning to reason, they still suffer from significant loss in reasoning capabilities during the transfer phase, when the model is required to adapt from perfectly clean visual input to the noisy input it will encounter after deployment. We conjecture, that reasoning on noisy data involves additional functional components not necessary in the clean case due to different types of domain shifts: (1) a *presence shift*, caused by imperfect object detectors, leading to missing visual objects necessary for

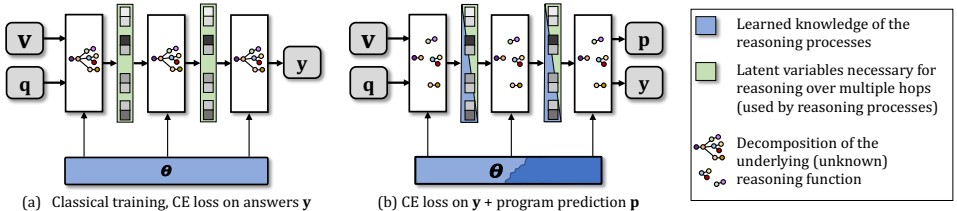

(a)  Classical training, CE loss on answers **y**    (b) CE loss on **y** + program prediction **p**

Figure 1: VQA takes visual input $v$ and a question $q$ and predicts a distribution over answers $y$. (a) Classical discriminative training encodes the full reasoning function in the network parameters $\theta$, while the network activations contain latent variables necessary for reasoning over multiple hops. (b) Additional program supervision requires intermediate network activations to contain information on the reasoning process, simplifying learning the reasoning function $g$. Under the hypothesis of it's decomposition into multiple reasoning modes, intermediate supervision favors separately learning the mode selector and each individual mode function. This intuition is analyzed theoretically in section 4.

reasoning, or to multiple (duplicate) detections; (2) an *appearance shift* causing variations in object embeddings (descriptors) for the same class of objects due to different appearance.

In this paper, we propose a new method for transferring reasoning patterns from models learned on perfect visual input to models trained on noisy visual representations. Key to the success is a regularization term minimizing loss of the reasoning capabilities during transfer. In particular, we address this problem through program prediction as an additional auxiliary loss, i.e. supervision of the sequence of reasoning operations along with their textual and/or visual arguments. To maintain a strong link between the learned function and its objective during the knowledge transfer phase, when inputs are switched from clean oracle inputs to noisy input, the neural model is required to continue to predict complex reasoning programs from different types of inputs.

As a second justification, we claim that program supervision in itself leads to a simpler learning problem, as the underlying reasoning function is decomposed into a set of tasks, each of which is easier to learn individually than the full joint decision function. We backup this claim through a theoretical analysis showing decreased sample complexity under mild hypotheses.

As a summary, we present the following contributions: **(i)** we propose a new program supervision module added on top of vision-language transformer models; **(ii)** we provide a theoretical analysis of the benefit of supervising program prediction in VQA deriving bounds on sample complexity; **(iii)** we experimentally demonstrate the efficiency of program supervision and show that it increases VQA performance on both in- and out-of-distribution sets, even when combined with BERT-like pre-training [30, 10], and that it improves the quality of oracle transfer initially proposed by [21].

## 2   Related Work

**Transformers in VQA** — VQA as a task was introduced in various datasets [4, 12, 16], including GQA [15] which is automatically-generated from real-world images. This growing amount of diverse datasets has been accompanied by the development of more and more sophisticated VQA models. While their exhaustive survey is out of the scope of this paper, one can mention some foundational categories of approaches, e.g. those based on object-level attention [2] and tensor decomposition [5]. In this work, we focus on VQA models which are based on Transformers [33], due to their wide adoption and their impressive results in several tasks (including VQA). In particular, we rely on the combination of Transformers with a large-scale BERT [10]-like pretraining which was shown to be beneficial for VQA in recent works [19, 30]. More recently, [21] focused on the study of so-called reasoning patterns in such Transformer-based VQA models. The authors analyzed how various VQA tasks are encoded in different attention heads, by applying an energy-based analysis inspired by [24]. The analysis was performed using a perfect-sighted oracle Transformer model (which is trained with near-to-perfect visual information, and thus, is much less prone to exploit dataset biases) in order to identify which patterns lead to better reasoning. Then, the authors showed that these reasoning patterns can be transferred from the oracle to a Transformer-based VQA model, thus improving both overall accuracy and accuracy on infrequent answers. In this work, we argue that using program prediction as an additional supervision signal is a catalyst for the transfer of these reasoning patterns.

**Biases and shortcut learning in VQA** — In addition to the popular VQA datasets mentioned above, other benchmarks were proposed to evaluate specific reasoning capabilities of VQA systems. In particular, they address the issue of shortcut learning [11] in deep learning, where models learn decision rules which are ineffective when tested on a domain or distribution different from the training one. For instance, VQA-CP [1] explicitly inverts the answer distribution between train and test splits. To cope with recent criticisms regarding these evaluations [31, 28], the GQA-OOD dataset [18] introduced a new split of GQA focusing on rare (Out-Of-Distribution / OOD) question-answer pairs, and showed that many VQA models strongly rely on dataset biases. Following this work, the VQA-CE dataset [8] introduced a new evaluation approach related to the VQA v2 dataset. By studying the multi-modal shortcuts in the training set and mining some trivial predictive rules (e.g. co-occurrences of words and visual objects), the evaluation set is generated using questions where these mined rules lead to incorrect answers. As for GQA-OOD, this dataset demonstrated that SOTA models do not perform well when they can't rely on shortcuts, even models which use bias-reduction techniques. Based on these benchmarks, several methods have been conceived to reduce shortcut learning in VQA (see [31, 29] for a comprehensive study of the different techniques). However none of them achieve significant performance improvement on GQA-OOD [18] or VQA-CE [8].

**Connections with symbolic VQA** — Our work is also related to neuro-symbolic reasoning [35, 23] and neural module networks (NMN) [3, 13], which generally encode a set of pre-defined functions into unique neural modules, then dynamically compose them to execute question-related programs. Several improvements of standard NMNs have been proposed to make them end-to-end trainable through reinforcement learning [17] or, more recently, to enhance their scalability and generalizability [6]. However, the common point of all these works is that they are generally based on the prediction of reasoning programs whose elementary functions are learned jointly with program prediction itself, through program supervision. In contrast to this work, our approach uses program supervision to enrich its internal representations instead of inferring program's execution.

**Measuring complexity of learning problems** — and thus generalization, has been a goal of theoretical machine learning since the early days, with a large body of work based on PAC-Learning [32, 27]. Traditionally, bounds have been provided ignoring data distributions and focusing uniquely on hypothesis classes (network structures in neural network language), e.g. as measured by VC-dimension. Surprising experimental results on training networks on random samples have seemingly contradicted learning theory [37], in particular Rademacher Complexity. Recently, building on statistics of gradient descent, bounds have been proposed which take into account data distributions, notably [26]. Algorithmic alignment between neural network structures and the decomposition of underlying reasoning functions has been studied in [34], with a focus on algorithms based on dynamic programming. Our theoretical contribution in section 4 builds on the latter two methodologies and extends this type of analysis to intermediate supervision of reasoning programs.

# 3    Knowledge transfer and program supervision

We propose a method for transferring reasoning patterns from models learned on perfect visual inputs to models trained on noisy visual representations. In the lines of *oracle transfer* [21], we first pre-train a model on ground-truth visual data and then fine-tune on standard visual embeddings extracted with an object detector. The underlying hypothesis is that the noise and uncertainty in visual input prevents the model from learning to reason and leads to learning shortcuts.

To minimize the loss of reasoning capabilities during the transfer over the *presence shift* between oracle and noisy visual objects, we propose a regularization technique, which supervises the prediction of reasoning steps required to answer the question. We therefore assume the existence of the following ground truth annotation of reasoning programs. A given data sample consists of a sequence $\{q_i\}$ of input question word embeddings, a set $\{v_i\}$ of input visual objects, the ground truth answer class $y^*$ as well as the groundtruth reasoning program, which is structured as a tree involving operations and arguments. Operations $\{o_i^*\}$ are elements of a predefined set {choose color, filter size, ...}. The arguments of these operations may be taken from (i) all question words, (ii) all visual objects, (iii) all operations — when an operation takes as argument the result of another operation. Hence, arguments are annotated as many-to-many relationships. In the question "*Is there a motorbike or a plane?*", for instance, the operation "or" depends on the result of the two operations checking the existence of a specific object in the image. This is denoted as $a_{ij}^{q*} \in \{0, 1\}$ where $a_{ij}^{q*} = 1$ means that

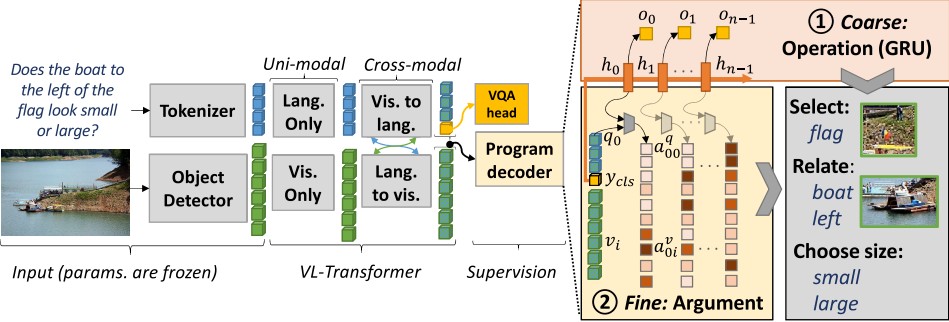

Figure 2: A vision+language transformer with an attached program decoder. The decoder is fed with the VL-Transformer's penultimate embedding (just before the VQA classification head) and generates programs using a coarse-to-fine approach: ① a coarse program is generated using a GRU, consisting of a sequence of program operations embeddings $\{o_i\}_{i\in[0,n-1]}$. ② It is then re-fined by predicting the visual $a_{ij}^v$ and textual $a_{ij}^q$ arguments using an affinity score between operation and input embeddings. Not shown: prediction of the operation's dependencies.

operation $i$ is associated with question word $j$ as argument and, similarly, $a_{ij}^{v*}{=}1$ indicating a visual argument and $a_{ij}^{d*}{=}1$ an operation result argument.

We propose to apply the regularization on top of the VL-Transformer architecture proposed in [30], based on sequences of self- and cross-modality attention. For this purpose, we define a trainable module for program generation (*program decoder*), added to the output of the VL-Transformer model as shown in Fig. 2 — an adaptation to other architectures would be straightforward.

**Program decoder** — In the lines of [6], the program decoder has been designed in a coarse-to-fine fashion. It first generates ① a coarse sketch of the program consisting only of the operations, which are then ② refined by predicting textual and visual arguments and dependencies between operations.

① **Coarse: operation** — this module only predicts the sequence of operations $\{o_i\}_{i\in[0,n-1]}$ using a recurrent neural network (RNN / GRU) [7] variant, whose initial hidden state is initialized with the $y_{CLS}$ token embedding of the VQA transformer — the same embedding from which classically the final answer $y$ is predicted, *cf.* Figure 2. Inference is stopped when the special "*STOP*" operation is predicted. At each GRU time step $i$, a new hidden state $h_i$ is computed, from which the operation $o_i$ is classified with a linear projection. It is supervised with a cross-entropy loss $\mathcal{L}_{op} = \sum_i \mathcal{L}_{CE}(o_i, o_i^*)$.

② **Fine: input arguments** — the coarse program is then refined by predicting the operations' arguments. We first deal with textual and visual arguments only. Affinity scores $a_{ij}^q$ between each operation's hidden $h_i$ and token $q_j$ embeddings are computed with a 2-layer feed-forward network from concatenated embeddings. They represent the probability of the word $q_j$ to belong to the argument set of operation $o_i$. Similar scores $a_{ij}^v$ are computed for operations and visual objects. They are supervised with BCE losses $\mathcal{L}_{qarg} = \sum_{ij}\mathcal{L}_{BCE}(a_{ij}^q, a_{ij}^{q*})$ and $\mathcal{L}_{varg} = \sum_{ij}\mathcal{L}_{BCE}(a_{ij}^v, a_{ij}^{v*})$.

**Fine: op arguments** — next the dependencies are predicted, i.e. arguments which correspond to results of other operations, and which structure the program into a tree. We deal with these arguments differently, and compute the set of dependency arguments for each operation $o_i$ with another GRU, whose hidden state is initialized with the hidden state $h_i$ of the operation. The argument index $a_{ij}^d$ is a linear projection of the hidden state and supervised with BCE: $\mathcal{L}_{dep} = \sum_{ij}\mathcal{L}_{BCE}(a_{ij}^d, a_{ij}^{d*})$.

**Program supervision** — The coarse-to-fine program decoder is trained with the four additional losses weighted by hyperparameters $\alpha, \beta, \gamma, \delta$.

$$\mathcal{L} = \underbrace{\mathcal{L}_{vqa}}_{\text{VQA}} + \underbrace{\alpha.\mathcal{L}_{op} + \beta.\mathcal{L}_{dep} + \gamma.\mathcal{L}_{qarg} + \delta.\mathcal{L}_{varg}}_{\text{Program supervision}}, \tag{1}$$

**Ground truth programs** — We use ground truth information from the GQA [15] dataset, whose questions have been automatically generated from real images. Each sample contains a program describing the operations and arguments required to derive the answer for each question. However, the GT programs have been created for GT visual arguments (GT objects), which do not exactly

match the visual input of an object detector used during training and inference [2]. We therefore construct a soft target, by computing intersection-over-union (IoU) between GT and detected objects.

**Oracle transfer** — Our method uses program supervision to regularize knowledge transfer from a visual oracle to noisy input, as introduced in [21]. Oracle transfer consists in pretraining the VL-Transformer model on ground-truth one-hot visual inputs before performing BERT-like pre-training. It offers training conditions which are more favorable for learning reasoning capabilities. We perform the following steps: **(1)** Oracle pre-training on GT visual input on the GQA dataset; **(2)** (optionally) BERT-like pre-training on data from GQA *unbalanced*; **(3)** Finetuning on the final VQA-objective on the GQA dataset. Each one of these steps are regularized using program supervision.

## 4 Sample complexity of program supervision

We provide a theoretical analysis indicating that the prediction and supervision of reasoning programs can improve learnability in vision and language reasoning under some assumptions. In what follows, we denote with $g$ "true" (but unknown) underlying reasoning functions, and by $f$ functions approximating them, implemented as neural networks. The goal is to learn a function $g$ able to predict a distribution $\boldsymbol{y}$ over answer classes given an input question and an input image, see Fig 1a. While in the experimental part we use state-of-the-art tranformer based models, in this theoretical analysis, we consider a simplified model, which takes as input the two vectorial embeddings $\boldsymbol{q}$ and $\boldsymbol{v}$ corresponding to, respectively, the question and the visual information (image), for instance generated by a language model and a convolutional neural network, and produces answers $\boldsymbol{y}^* = g(\boldsymbol{q}, \boldsymbol{v})$.

We restrict this analysis to two-layer MLPs, as they are easier to handle theoretically than modern attention based models. The reasoning function $g$ is approximated by neural network $f$ parametrized by a vector $\theta$ and which predicts output answers $\boldsymbol{y} = f(\boldsymbol{q}, \boldsymbol{v}, \theta)$.

Our analysis uses PAC-learning [32] and builds on recent results providing bounds on sample complexity taking into account the data distribution itself. We here briefly reproduce Theorem 3.5. from paper [34], which, as an extension of a result in [26], provides a bound for sample complexity of overparametrized MLPs with vectorial outputs, i.e. MLPs with sufficient capacity for learning a given task:

**Theorem 4.1** (Sample complexity for overparametrized MLPs). *Let $\mathcal{A}$ be an overparametrized and randomly initialized two-layer MLP trained with gradient descent for a sufficient number of iterations. Suppose $g : \mathbb{R}^d \to \mathbb{R}^m$ with components $g(x)^{(i)} = \sum_j \alpha_j^{(i)} (\beta_j^{(i)T} x)^{p_j^{(i)}}$, where $\beta_j^{(i)} \in \mathbb{R}^d$, $\alpha^{(i)} \in \mathbb{R}$, and $p_j^{(i)} = 1$ or $p_j^{(i)} = 2l, l \in \mathbb{N}_+$. The sample complexity $\mathcal{C}_{\mathcal{A}}(g, \epsilon, \delta)$ is*

$$\mathcal{C}_{\mathcal{A}}(g, \epsilon, \delta) = O\left( \frac{\max_i \sum_j p_j^{(i)} |\alpha_j^{(i)}| \cdot ||\beta_j^{(i)}||_2^{p_j^{(i)}} + \log(\frac{m}{\delta})}{(\epsilon/m)^2} \right) \tag{2}$$

We use the following *Ansatz*: since each possible input question requires a potentially different form of reasoning over the visual content, our analysis is based on the following assumption.

**Assumption 1.** *The unknown reasoning function $g()$ is a mixture model which decomposes as follows*

$$\boldsymbol{y}^* = \sum_r \boldsymbol{\pi}_r \boldsymbol{h}_r = \sum_r \boldsymbol{\pi}_r g_r(\boldsymbol{v}), \tag{3}$$

*where the different mixture components $r$ correspond to different forms of reasoning related to different questions. The mixture components can reason on the visual input only, and the mixture weights are determined by the question $\boldsymbol{q}$, i.e. the weights $\boldsymbol{\pi}$ depend on the question $\boldsymbol{q}$, e.g. $\boldsymbol{\pi} = g_\pi(\boldsymbol{q})$.*

We call $g_\pi(.)$ the *reasoning mode estimator*. One hypothesis underlying this analysis is that learning to predict reasoning programs allows the model to more easily decompose into this form (3), i.e. that the network structure closely mimics this decomposition, as information on the different reasoning modes $r$ is likely to be available in the activations of intermediate layers, *cf.* Figure 1. This will be formalized in assumption 3 and justified further below.

Considering the supposed "true" reasoning function $\boldsymbol{y}^* = g(\boldsymbol{q}, \boldsymbol{v})$ and its decomposition given in (3), we suppose that each individual reasoning module $g_r$ can be approximated with a multi-variate

polynomial, in particular each component $\boldsymbol{h}_r^{(i)}$ of the vector $\boldsymbol{h}_r$, as

$$\boldsymbol{h}_r^{(i)} = g_r(\boldsymbol{v}) = \sum_j \alpha_{r,j}^{(i)}(\beta_{r,j}^{(i)T}\boldsymbol{v})^{p_{r,j}^{(i)}} \quad \text{with parameters} \quad \omega = \left\{\alpha_{r,j}^{(i)}, \beta_{r,j}^{(i)}, p_{r,j}^{(i)}.\right\} \tag{4}$$

A trivial lower bound on the complexity of the reasoning mode estimator $g_\pi(.)$ is the complexity of the identity function, which is obtained in the highly unlikely case where the question embeddings $\boldsymbol{q}$ contain the 1-in-K encoding of the choice of reasoning mode $r$. We adopt a more realistic case as the following assumption.

**Assumption 2.** *The input question embeddings $\boldsymbol{q}$ are separated into clusters according to reasoning modes $r$, such that the underlying reasoning mode estimator $g_\pi$ can be realized as a NN classifier with dot-product similarity in this embedding space.*

Under this assumption, the reasoning mode estimator can be expressed as a generalized linear model, i.e. a linear function followed by a soft-max $\sigma$,

$$\boldsymbol{\pi} = g_\pi(\boldsymbol{q}) = \sigma\left(\left[\gamma_0^T\boldsymbol{q}, \ \gamma_1^T\boldsymbol{q}, ...\right]\right), \tag{5}$$

where the different $\gamma_r$ are the cluster centers of the different reasoning modes $r$ in the question embedding space. As the softmax is a monotonic non-linear function, its removal will not decrease sample complexity[1], and the complexity can be bounded by the logits $\boldsymbol{\pi}_r = \gamma_r^T\boldsymbol{q}$. Plugging this into (3) we obtain that each component $\boldsymbol{y}^{*(i)}$ of the answer is expressed as the following function:

$$\boldsymbol{y}^{*(i)} = \sum_r \left(\gamma_r^T\boldsymbol{q}\right) \sum_j \alpha_{r,j}^{(i)}(\beta_{r,j}^{(i)T}\boldsymbol{v})^{p_{r,j}^{(i)}} \tag{6}$$

We can reparametrize this function by concatenating the question $\boldsymbol{q}$ and the visual input $\boldsymbol{v}$ into a single input vector $\boldsymbol{x}$, which are then masked by two different binary masks, which can be subsumed into the parameters $\gamma_r$ and $\beta_{r,j}^{(i)}$, respectively, giving

$$\boldsymbol{y}^{*(i)} = \sum_r \sum_j (\gamma_r^T\boldsymbol{x})\alpha_{r,j}^{(i)}(\beta_{r,j}^{(i)T}\boldsymbol{x})^{p_{r,j}^{(i)}} \tag{7}$$

Extending Theorem 3.5. from [34], we can give our main theoretical result as the sample complexity of this function expressed as the following theorem.

**Theorem 4.2** (Sample complexity for multi-mode reasoning functions). *Let $\mathcal{A}$ be an over-parametrized and randomly initialized two-layer MLP trained with gradient descent for a sufficient number of iterations. Suppose $g : \mathbb{R}^d \to \mathbb{R}^m$ with components $g(x)^{(i)} = \sum_r \sum_j (\gamma_r^T\boldsymbol{x})\alpha_{r,j}^{(i)}(\beta_{r,j}^{(i)T}\boldsymbol{x})^{p_{r,j}^{(i)}}$ where $\gamma_r \in \mathbb{R}^d$, $\beta_{r,j}^{(i)} \in \mathbb{R}^d$, $\alpha_{r,j}^{(i)} \in \mathbb{R}$, and $p_{r,j}^{(i)} = 1$ or $p_{r,j}^{(i)} = 2l, l \in \mathbb{N}_+$. The sample complexity $\mathcal{C}_\mathcal{A}(g, \epsilon, \delta)$ is*

$$\mathcal{C}_\mathcal{A}(g, \epsilon, \delta) = O\left(\frac{\max_i \sum_r \sum_j \pi p_{r,j}^{(i)}|\alpha|\cdot||\gamma_r||_2\cdot||\beta_{r,j}||_2^{p_{r,j}^{(i)}} + \log(m/\delta)}{(\epsilon/m)^2}\right).$$

The proof of this theorem is given in the supplementary material (Appendix A).

Theorem 4.2 provides the sample complexity of the reasoning function $g()$ under classical training. In the case of program supervision, our analysis is based on the following assumption (see also Fig. 1b):

**Assumption 3.** *Supervising reasoning programs encodes the choice of reasoning modes $r$ into the hidden activations of the network $f$. Therefore, learning is separated into several different processes,*

  (a) *learning of the reasoning mode estimator $g_\pi()$ approximated as a network branch $f_\pi()$ connected to the program output;*

  (b) *learning of the the different reasoning modules $g_r()$ approximated as network branches $f_r()$ connected to the different answer classes $\boldsymbol{y}_r$; each one of these modules is learned independently.*

---

[1]In principle, there should exist special degenerate cases, where an additional softmax could reduce sample complexity; however, in our case it is applied to a linear function and thus generates a non-linear function.

| | Model | Oracle transf. | Prog. sup. | GQA-OOD [18] | | GQA [15] | | | | AUC† |
|---|---|---|---|---|---|---|---|---|---|---|
| | | | | acc-tail | acc-head | test-dev | binary* | open* | test-std | prog. |
| scratch | (a) Baseline | | | 42.9 | 49.5 | 52.4 | - | - | - | / |
| | (b) Oracle transfer | ✓ | | $48.2_{\pm0.3}$ | $54.6_{\pm1.1}$ | $57.0_{\pm0.3}$ | 74.5 | 42.1 | 57.3 | / |
| | **(c) Ours** | ✓ | ✓ | $\mathbf{48.8}_{\pm0.1}$ | $\mathbf{56.1}_{\pm0.3}$ | $\mathbf{57.8}_{\pm0.2}$ | **75.4** | **43.0** | **58.2** | 97.1 |
| + lxmert | (d) Baseline | | | 47.5 | 55.2 | 58.5 | - | - | - | / |
| | (e) Oracle transfer | ✓ | | 47.1 | 54.8 | 58.4 | 77.1 | 42.6 | 58.8 | / |
| | (f) **Ours** | ✓ | ✓ | $\mathbf{48.0}_{\pm0.6}$ | $\mathbf{56.6}_{\pm0.6}$ | $\mathbf{59.3}_{\pm0.3}$ | **77.3** | **44.1** | **59.7** | 96.4 |

Table 1: Impact of program supervision on *Oracle transfer* [21] for vision-language transformers. LXMERT [30] pre-training is done on the GQA unbalanced training set. We report scores on GQA [15] (*test-dev* and *test-std*) and GQA-OOD (*test*). * binary and open scores are computed on the test-std; † we evaluate visual argument prediction by computing AUC@0.66 on GQA-val.

We justify Assumption 3.a through supervision directly, which separates $g_\pi()$ from the rest of the reasoning process. We justify Assumption 3.b by the fact, that different reasoning modes $r$ will lead to different hidden activations of the network. Later layers will therefore see different inputs for different modes $r$, and selector neurons can identify responsible inputs for each branch $f_r()$, effectively switching off irrelevant input.

We can see that these complexities are lower than the sample complexity of the full reasoning function given in theorem 4.2, since for a given combination of $i, r, j$, the term $||\gamma_r||_2 \cdot ||\beta_{r,j}||_2^{p_{r,j}^{(i)}}$ dominates the corresponding term $||\beta_{r,j}||_2^{p_{r,j}^{(i)}}$. Recalling that the vectors $\gamma$ correspond to the cluster centers of reasoning modes in language embedding space. Under the assumption that the embeddings $q$ have been created with batch normalization, a standard technique in deep learning, each value $\gamma_r^{(i)}$ follows a normal distribution $\mathcal{N}(0, 1)$. Dropping indices $i, r, j$ to ease notation, we can then compare the expectation of the term $||\gamma||_2 \cdot ||\beta||_2^p$ over the distribution of $\gamma$ and derive the following relationship:

$$\mathbb{E}_{\gamma^{(i)} \sim N(0,1)} ||\gamma||_2 \cdot ||\beta||_2^p = C||\beta||_2^p = \sqrt{2} \frac{\Gamma(\frac{m}{2} + \frac{1}{2})}{\Gamma(\frac{m}{2})} ||\beta||_2^p \tag{8}$$

where $\Gamma$ is the Gamma special function and $m$ is the dimension of the language embedding $\gamma$. We provide a proof for this equality in the supplementary material (Appendix A).

**Discussion and validity of our claims** — the difference in sample complexity is determined by the factor $C$ in equation (8), which monotonically grows with the size of the embedding space $m$, which is typically in the hundreds. For the order of $m=512$ to $m=768$ used for state-of-the-art LXMERT models [30], complexity grows by a factor of around $\sim20$. We would like to point out that this analysis very probably under-estimates the difference in complexity, as the difference highly depends on the complexity of the reasoning estimator $\pi$, which we have simplified as a linear function in equation (5). Taking into account just the necessary soft-max alone would probably better appreciate the difference in complexity between the two methods, which we leave for future work. Our analysis is also based on several assumptions, among which is the simplified model (an over-parametrized MLP instead of an attention based network), as well as assumptions of Theorem 4.2 from [34] and [26], on which our analysis is based. Lastly, we would like to comment on the fact that we compare two different bounds: (i) the one on sample complexity for learning the full multi-modal reasoning given in Theorem 4.2, and (ii) the one for learning a single reasoning mode given by Theorem 4.1. While comparing bounds does not provide definitive answers on the order of models, both bounds have been derived by the same algebraic manipulations and we claim that they are comparable.

## 5   Experimental results

**Setup + architecture:** — we perform our experiments with a compact version of the Vision-Language (VL)-Tansformer used in [30] (cf. Fig. 2), with a hidden embedding size of $d=128$ and $h=4$ heads per layer (only 26M trainable parameters). **Dataset:** our models are trained on the balanced GQA [15] training set ($\sim$1M question-answer pairs). However, LXMERT pretraining is done on the *unbalanced* training set ($\sim$15M question-answer pairs). The latter contains more questions and programs, but the same number of images ($\sim$100K images). **Evaluation:** is performed on GQA [15]

| Ablations | GQA-OOD [18] acc-tail (val.) | GQA [15] val. |
|---|---|---|
| (1) VQA only | 46.9 | 62.2 |
| (2) Coarse only | 46.5 | 62.5 |
| (3) Coarse + dep. | 46.8 | 62.8 |
| (4) Full w/o v.arg | 47.3 | 63.7 |
| **(5) Full (ours)** | **49.9** | **66.2** |

Table 2: Ablation of different types of program supervision (compact model, no LXMERT/BERT pre-training, no Oracle), on GQA val. *v.arg* = superv. of visual arguments.

| Ablations | GQA-OOD [18] acc-tail (val.) | GQA [15] val. |
|---|---|---|
| (6) No prog | 50.0 | 66.4 |
| (7) Uni-modal | 49.9 | 66.5 |
| **(8) Cross-modal** | **50.4** | **67.4** |

Table 3: We study the impact of the program supervision posistion: after uni-modal layers or after cross-modal layers (*standard configuration*). The supervision is more efficient when used after cross-modal interactions. Setting=oracle transfer, no lxmert

and GQA-OOD [18] test sets. GQA is a dataset with question-answer pairs automatically generated from real images, and is particularly well suited for evaluating a large variety of reasoning skills. GQA-OOD is a benchmark dedicated to the out-of-domain VQA evaluation, and gives information on the generalization capabilities of the model. Hyper-parameters are selected either on the testdev (for GQA) or validation (for GQA-OOD) sets. When specified (with ±) we provide the average accuracy and standard deviation computed on three runs with different random seeds. **Visual input:** following [2], we use bottom-up visual features extracted using a pre-trained object detector (we keep its parameters frozen during the training). If not specified, we use faster-RCNN [25] with 36 objects per-images. In addition to that, we experiment with designed objects, and also with the VinVL [38] features which (unlike faster-RCNN ones) are conceived specifically for vision-language tasks.

**Program supervision improves visual reasoning** — Tab. 1 reports the effectiveness of program prediction when combined with oracle and BERT-like pretraining on the GQA dataset and corroborates the results found in the theoretical analysis. In addition, when using both program supervision and LXMERT [30] but without oracle transfer, we achieve an accuracy of $58.8$ on the *testdev* set of GQA. This is lower than oracle transfer's accuracy, demonstrating the complementary of the two methods. We note that the majority of the gain is achieved on the more challenging *open* questions. In addition, results on GQA-OOD (*acc-tail* and *acc-head*) suggest that the gains are obtained in, both, out- and in-distribution settings. However, as already observed in [21], LXMERT pre-training tends to decrease the *acc-tail* gains brought by oracle transfer plus program supervision. We evaluate the program prediction performance by measuring the area under the ROC curve (AUC) on the visual argument prediction with an IoU treshold of $\frac{2}{3}$=0.66. Models (c) and (e) achieve, respectively, $97.1$ and $96.4$ AUC scores, demonstrating the effectiveness of the program decoder.

**Program supervision decreases sample complexity** — as shown in Fig. 3. We vary the amount of training data from $5\%$ to $100\%$ and observe that adding program supervision allows to reach an accuracy similar to the baseline while using less data (no *oracle transfer*, no LXMERT pretraining).

**Visual arguments are the key** — We study the impact of different types of program supervision in Tab. 2. We can see the importance of supervising arguments, in (4) and (5). The supervision of visual arguments (5) contributes most to the gain in performance, again corroborating that visual uncertainty is the main bottleneck for reasoning.

**Program supervision enhances cross-modal interactions** — In Tab. 3, we study how the inputs of the program prediction module influence the VQA accuracy. In particular, we test two settings: (7) *uni-modal*, where the programs are predicted from the vision and language em-

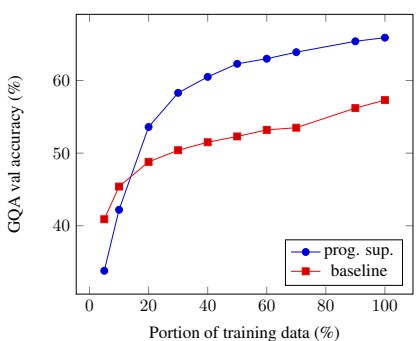

Figure 3: Empirical evaluation of sample complexity of program supervision.

beddings right after the uni-modal layers (language and vision only in Fig. 2); and (8) *cross-modal*, where the programs are predicted after the cross-modal layers (as shown in Fig. 2). We observe that, contrary to the latter, the former does not improve the baseline ((7) vs (6) in Tab. 3). This highlights the fact that the program supervision mainly impacts the operations in the cross modal layers, where the most complex reasoning operations are performed.

| Model | Visual features | Oracle transf. | Prog. sup. | GQA [15] | | | |
|---|---|---|---|---|---|---|---|
| | | | | test-dev | binary* | open* | test-std |
| (g) Oracle transfer | 100 RCNN [2] | ✓ | | $57.0_{\pm 0.4}$ | - | - | - |
| **(h) Ours** | | ✓ | ✓ | $\mathbf{58.2}_{\pm 0.1}$ | - | - | - |
| (i) Oracle transfer | VinVL [38] | ✓ | | $59.6_{\pm 0.1}$ | - | - | - |
| **(j) Ours** | | ✓ | ✓ | $\mathbf{60.9}_{\pm 0.2}$ | - | - | - |
| (k) Oracle transfer +lxmert | | ✓ | | 61.4 | 79.6 | 47.5 | 62.5 |
| **(l) Ours** +lxmert | | ✓ | ✓ | **61.8** | **80.1** | **48.0** | **63.0** |

Table 4: Impact of improved visual inputs while using program supervision on Vision-Language Transformers. Scores on GQA [15]. *binary/open are computed on test-std.

| Method | Visual feats. | Additional supervision | Training data (M) | | GQA-OOD [18] | | GQA [15] | | |
|---|---|---|---|---|---|---|---|---|---|
| | | | Img | Sent | acc-tail | acc-head | bin. | open | all |
| BAN4 [22] | RCNN [2] | - | $\approx 0.1$ | $\approx 1$ | 47.2 | 51.9 | 76.0 | 40.4 | 57.1 |
| MCAN [36] | RCNN [2] | - | $\approx 0.1$ | $\approx 1$ | 46.5 | 53.4 | 75.9 | 42.2 | 58.0 |
| Oracle transfer [21] | RCNN [2] | - | $\approx 0.18$ | $\approx 1$ | 48.3 | 55.5 | 75.2 | 44.1 | 58.7 |
| MMN [6] | RCNN [2] | Program | $\approx 0.1$ | $\approx 15$ | 48.0 | 55.5 | 78.9 | 44.9 | 60.8 |
| LXMERT [30] | RCNN [2] | - | $\approx 0.18$ | $\approx 9$ | **49.8** | 57.7 | 77.8 | 45.0 | 60.3 |
| **Ours** | VinVL [38] | Program | $\approx 0.1$ | $\approx 15$ | 49.1 | **59.7** | 80.1 | 48.0 | 63.0 |
| NSM [14] | SG [14] | Scene graph | $\approx 0.1$ | $\approx 1$ | - | - | 78.9 | **49.3** | 63.2 |
| OSCAR+VinVL [38] | VinVL [38] | - | $\approx 5.7$ | $\approx 9$ | - | - | **82.3** | 48.8 | **64.7** |

Table 5: Comparison with the state of the art on the GQA [15] (*test-std*) and GQA-OOD [18] (*test*) sets. For a fair comparison, we provide information about the required training data and supervision.

**Program supervision allows to take advantage of the improved visual inputs** — We analyse the impact of using our method with a better input image representation. Increasing the number of objects from 36 to 100 per image ((g) and (h) in Tab. 4), allows to further increase the gains brought by our method. On the contrary, the score of the baseline model remains unchanged, showing that the program supervision allows to take advantage of a bigger number of object proposals. Similarly, replacing the faster-RCNN features by the more recent and more accurate VinVL ones ((i)-(l) in Tab. 4) results in better performances.

**Comparison with SOTA** — We report in Tab. 5 the results obtained by our approach compared to the current SOTA on the GQA and GQA-OOD datasets. In order to ensure a fair comparison, we also provide, for each method, information regarding the amount of data (images and sentences) used during training. As shown in Tab. 5, our approach compares favourably with SOTA since it obtains the second best accuracy (with a 0.2 points gap) on the GQA test-std set among the approaches which not use extra training data. The results also remain competitive when comparing to the OSCAR+VinVL [38], while being trained with 50 times less images. On GQA-OOD, our approach obtains the second best *acc-tail* score (and the best *acc-head* one) with a much less complex architecture than current SOTA (26M vs 212M trainable parameters compared to LXMERT [30]).

## 6  Conclusion

We have demonstrated that it is possible to improve the reasoning abilities of VQA models when providing additional supervision of program annotations. In particular, our method is designed to improve the transfer of reasoning patterns from models learned on perfect visual input to models trained on noisy visual representations. Our experiments are supported by a theoretical analysis, demonstrating that program supervision can decrease sample complexity under reasonable hypothesis. The proposed method relies on the availability of reasoning program annotations, which are costly to annotate, especially when dealing with human generated questions. Recent work has already managed to gather such kind of annotations [9]. The next step will be to extend the method to configurations where the program annotation is rare or incomplete. Another promising research path could be explicitly conditioning the VQA system's answer by the program prediction module, following the recent work of Chen et al. [6], instead of using it as catalyst for knowledge transfer.

**Acknowledgment** — C. Wolf acknowledges support from ANR through grant "*Remember*" (ANR-20-CHIA-0018).

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

**Broader Impact** — Beyond the exciting scientific reasons for exploring work on visual reasoning, we welcome the potentially high interest for society in VQA systems targeting increased accessibility. As an example, helping the visually impaired to query their environment based on a camera input. We are currently not aware of existing applications abusing VQA systems for unethical goals. Potential future unethical abuse could involve their use to automatic solving of Captchas (automatic Turing tests), and the creation of false dialogs and discussions in online forums ("troll farms") requiring the examination of posted text as well as accompanying images. The latter is a risk inherent to any powerful system capable of predicting text, but for the moment does not realistically apply to VQA, where very short answers are predicted through classification over the full answer dictionary (even though our contribution is independent of the chosen VQA model).

