# Supervising the Transfer
# of Reasoning Patterns in VQA

Corentin Kervadec*[1,2]    Christian Wolf*[2]    Grigory Antipov[1]    Moez Baccouche[1]
Madiha Nadri[3]

[1]Orange Innovation, France    [2]LIRIS, INSA-Lyon, France    [3]LAGEPP, Université de Lyon, France
firstname.lastname@orange.com, christian.wolf@insa-lyon.fr,
madiha.nadri@lagep.univ-lyon1.fr, *equal contribution

# Supplementary Material

## A    Proofs of Section 4

### A.1    Proof of theorem 4.2

For the unfamiliar reader, we here briefly recall the notion of sample complexity, in the context of PAC-learning [14], which characterizes the minimum amount ($=M$) of samples necessary to learn a function with sufficiently low ($=\epsilon$) error with sufficiently high ($=\delta$) probability:

**Definition A.1** (Sample complexity). Given an error threshold $\epsilon>0$; a threshold on error probability $\delta$; a training set $S = \{x_i, y_i\}$ of $M$ i.i.d. training samples from $\mathcal{D}$, generated from some underlying true function $y_i = g(\boldsymbol{x}_i)$, and a learning algorithm $\mathcal{A}$, which generates a function $f$ from training data, e.g. $f = \mathcal{A}(S)$; Then g is $(M, \epsilon, \delta)$-learnable by $\mathcal{A}$ if

$$\mathbb{P}_{x \sim D}\left[||f(x) - g(x)|| \leq \epsilon\right] \geq 1 - \delta \tag{1}$$

**The case of scalar outputs**    In the lines of [11], we first define the case for a single component $\boldsymbol{y}^{(i)}$ of the vector $\boldsymbol{y}$ and define the following Corollary:

**Corollary 0.1** (Sample complexity for multi-mode reasoning functions with a single scalar component). *Let $\mathcal{A}$ be an overparametrized and randomly initialized two-layer MLP trained with gradient descent for a sufficient number of iterations. Suppose $g : \mathbb{R}^d \rightarrow \mathbb{R}^m$ with $g(x) = \sum_r \sum_j (\gamma_r^T \boldsymbol{x}) \alpha_{r,j} (\beta_{r,j}^T \boldsymbol{x})^{p_{r,j}}$ where $\gamma_r \in \mathbb{R}^d$, $\beta_{r,j} \in \mathbb{R}^d$, $\alpha_{r,j} \in \mathbb{R}$, and $p_{r,j} = 1$ or $p_{r,j} = 2l, l \in \mathbb{N}_+$. The sample complexity $\mathcal{C}_{\mathcal{A}}(g, \epsilon, \delta)$ is*

$$C_A(g, \epsilon_0, \delta_0) =$$
$$O\left(\frac{\sum_r \sum_j \pi p_{r,j} |\alpha| \cdot ||\gamma_r||_2 \cdot ||\beta_{r,j}||_2^{p_{r,j}} + \log(\frac{1}{\delta_0})}{\epsilon_0^2}\right),$$

Proof of Corollary 0.1:

Using Theorem 5.1 from [11], we know that sums of learnable functions are learnable, and can thus focus on a single term

$$y = g(\boldsymbol{x}) = \alpha(\gamma^T \boldsymbol{x})(\beta^T \boldsymbol{x})^p \tag{2}$$

where we dropped indices $r$ and $j$ and the superscript $(i)$ for convenience.

We proceed in the lines of the proof of Theorem 5.1 in [11]. Given a set of i.i.d data samples $S = \{(\boldsymbol{x}_s, y_s)\}_{s=1}^n = (\boldsymbol{X}, \boldsymbol{y})$ from the underlying function $g(x)$, let $\boldsymbol{w}$ be the weights of the first

35th Conference on Neural Information Processing Systems (NeurIPS 2021).

layer of two layer network with ReLu activations; let $H^\infty \in \mathbb{R}^{n,n}$ be a Gram matrix defined as follows, with elements

$$H^\infty_{ij} = \mathbb{E}_{\boldsymbol{w} \sim \mathcal{N}(0,1)} \left[ \boldsymbol{x}_i^T \boldsymbol{x}_j \mathbb{I}\{\boldsymbol{w}^t \boldsymbol{x}_i {\geq} 0, \boldsymbol{w}^t \boldsymbol{x}_i {\geq} 0\} \right].$$

To provide bounds on the sample complexity of $g(x)$, using Theorem 5.1 of [11], it suffices to show that the following bound holds

$$\sqrt{\boldsymbol{y}^T (\boldsymbol{H}^\infty)^{-1} \boldsymbol{y}} < M_g \tag{3}$$

for a bound $M_g$ independent of the number of samples $n$.

For first introduce some notation. For matrices $\boldsymbol{A} = [\boldsymbol{a}_1, ..., \boldsymbol{a}_{n_3}] \in \mathbb{R}^{n_1 \times n_3}$ and $\boldsymbol{B} = [\boldsymbol{b}_1, ..., \boldsymbol{b}_{n_3}] \in \mathbb{R}^{n_2 \times n_3}$, the *Khatri-Rao* product is defined as $\boldsymbol{A} \odot \boldsymbol{B} = [\boldsymbol{a}_1 \otimes \boldsymbol{b}_1, \boldsymbol{a}_2 \otimes \boldsymbol{b}_2, ..., \boldsymbol{a}_{n_3} \otimes \boldsymbol{b}_{n_3}]$. Let $\circ$ be the *Haddamard* product (element wise multiplication) of two matrices. We also denote the corresponding powers by $\boldsymbol{A}^{\otimes l}, \boldsymbol{A}^{\odot l}, \boldsymbol{A}^{\circ l}$. We denote by $\boldsymbol{A}^\dagger = (\boldsymbol{A}^T \boldsymbol{A})^{-1} \boldsymbol{A}^T$ the *Moore-Penrose* pseudo-inverse, and by $\boldsymbol{P}_{\boldsymbol{A}} = \boldsymbol{A}^{\frac{1}{2}} \boldsymbol{A}^\dagger \boldsymbol{A}^{\frac{1}{2}}$ the projection matrix for the subspace spanned by $\boldsymbol{A}$.

From the proof of Theorem 5.1 in [11], we also know that

$$\boldsymbol{H}^\infty \succeq \frac{\boldsymbol{K}^{\circ 2l}}{2\pi(2l-1)^2},$$

where $\boldsymbol{K} = \boldsymbol{X}^T \boldsymbol{X}$, and $\boldsymbol{X}$ is the data matrix of all row vectors $\boldsymbol{x}_i$.

Let us consider the case of $p = 1$. Reformulating equation (2), we get:

$$y = g(\boldsymbol{x}) = \alpha(\gamma^T \boldsymbol{x})(\beta^T \boldsymbol{x}) \tag{4}$$
$$= \alpha(\boldsymbol{x}^T \gamma)(\boldsymbol{x}^T \beta) \tag{5}$$
$$= \alpha(\boldsymbol{x} \otimes \boldsymbol{x})^T (\gamma \otimes \beta) \tag{6}$$

Now, taking the full set of input vectors $\boldsymbol{x}_i$ arranged into the full data matrix $\boldsymbol{X}$, we can perform similar algebraic operations to get

$$\boldsymbol{y} = g(\boldsymbol{X}) = \alpha(\boldsymbol{X}^T \gamma) \circ (\boldsymbol{X}^T \beta) \tag{7}$$
$$= \alpha(\boldsymbol{X}^{\odot 2})^T (\gamma \otimes \beta) \tag{8}$$

Plugging (7) and (8) into (3), we need to show that the following expression is smaller than a constant $M_g$:

$$\alpha^2 ((\boldsymbol{X}^T \gamma) \circ (\boldsymbol{X}^T \beta))^T (\boldsymbol{H}^\infty)^{-1} (\boldsymbol{X}^{\odot 2})^T (\gamma \otimes \beta) \tag{9}$$
$$= \alpha^2 ((\boldsymbol{X}^{\odot 2})^T (\gamma \otimes \beta))^T (\boldsymbol{H}^\infty)^{-1} (\boldsymbol{X}^{\odot 2})^T (\gamma \otimes \beta) \tag{10}$$
$$= \alpha^2 (\gamma \otimes \beta)^T (\boldsymbol{X}^{\odot 2})(\boldsymbol{H}^\infty)^{-1} (\boldsymbol{X}^{\odot 2})^T (\gamma \otimes \beta) \tag{11}$$
$$\leq 2\pi \alpha^2 (\gamma \otimes \beta)^T (\boldsymbol{X}^{\odot 2})(\boldsymbol{K}^{\circ 2})^\dagger (\boldsymbol{X}^{\odot 2})^T (\gamma \otimes \beta) \tag{12}$$
$$= 2\pi \alpha^2 (\gamma \otimes \beta)^T \boldsymbol{P}_{\boldsymbol{X}^{\odot 2}(\boldsymbol{X}^{\odot 2})^T} (\gamma \otimes \beta) \tag{13}$$
$$\leq 2\pi \alpha^2 \|(\gamma \otimes \beta)\|_2^2 \tag{14}$$
$$= 2\pi \alpha^2 \|\gamma\|_2^2 \cdot \|\beta\|_2^2 \tag{15}$$

where we made use of $\|a \otimes b\|_2^2 = \|a\|_2^2 \|b\|_2^2$ for two vectors $a$ and $b$ and an integer $n$.

This finishes the proof for the case $p = 1$.

Let us consider the case of $p = 2l+1$. Reformulating equation (2), we get:

$$\boldsymbol{y} = g(\boldsymbol{X}) = \alpha(\boldsymbol{X}^T \gamma) \circ (\boldsymbol{X}^T \beta)^p \tag{16}$$
$$= \alpha(\boldsymbol{X}^{\odot 2l})^T (\gamma \otimes \beta^{\otimes(2l+1)}) \tag{17}$$

Plugging (17) into (3), we again need to show that the following expression is smaller than a constant $M_g$:

$$\alpha^2((\boldsymbol{X}^{\odot 2l})^T(\gamma \otimes \beta^{\otimes(2l+1)}))^T \tag{18}$$

$$(\boldsymbol{H}^\infty)^{-1}(\boldsymbol{X}^{\odot 2l})^T(\gamma \otimes \beta^{\otimes(2l+1)}) \tag{19}$$

$$=\alpha^2(\gamma \otimes \beta^{\otimes(2l+1)})^T \tag{20}$$

$$(\boldsymbol{X}^{\odot 2l})(\boldsymbol{H}^\infty)^{-1}(\boldsymbol{X}^{\odot 2l})^T(\gamma \otimes \beta^{\otimes(2l+1)}) \tag{21}$$

$$\leq 2\pi(2l-1)^2\alpha^2(\gamma \otimes \beta^{\otimes(2l+1)})^T \tag{22}$$

$$(\boldsymbol{X}^{\odot 2l})(\boldsymbol{K}^{\circ 2})^\dagger(\boldsymbol{X}^{\odot 2l})^T(\gamma \otimes \beta^{\otimes(2l+1)}) \tag{23}$$

$$=2\pi(2l-1)^2\alpha^2(\gamma \otimes \beta^{\otimes(2l+1)})^T \tag{24}$$

$$\boldsymbol{P}_{\boldsymbol{X}^{\odot 2l}(\boldsymbol{X}^{\odot 2l})^T}(\gamma \otimes \beta^{\otimes(2l+1)}) \tag{25}$$

$$\leq 2\pi(2l-1)^2\alpha^2||(\gamma \otimes \beta^{\otimes(2l+1)})||_2^2 \tag{26}$$

$$\leq 2\pi p^2\alpha^2||(\gamma \otimes \beta^{\otimes(2l+1)})||_2^2 \tag{27}$$

$$=2\pi p^2\alpha^2||\gamma||_2^2 \cdot ||\beta||_2^{2p} \tag{28}$$

where we made use of $||a \otimes b||_2^2 = ||a||_2^2||b||_2^2$ and therefore $||a^{\otimes n}||_2^2 = ||a||_2^{2n}$ for two vectors $a$ and $b$ and an integer $n$.

This finishes the proof for the case $p = 2l+1$.

**The case of vectorial outputs**  In the lines of [15], we consider each component of the output vector independent and apply an union bound to Corollary 0.1. If the individual components $\boldsymbol{y}^{(i)}$ fail to learn with probability $\delta_0$, then the full output of dimension $m$ fails with probability $m\delta_0$ and with an error of at most $m\epsilon_0$. A change of variables from $(\epsilon_0, \delta_0)$ to $(\epsilon, \delta)$ gives a complexity for the model with vectorial output of

$$\mathcal{C}_\mathcal{A}(g, \epsilon, \delta) =$$
$$O\left(\frac{\max_i \sum_r \sum_j \pi p_{r,j}^{(i)}|\alpha| \cdot ||\gamma||_2 \cdot ||\beta_{r,j}||_2^{p_{r,j}^{(i)}} + \log(m/\delta)}{(\epsilon/m)^2}\right),$$

This ends the proof of Theorem 4.2.

## A.2   Proof of the inequality in Eq. (8)

Let us denote by $p(x)$ the density of normal distribution. And to make the notation more succinct and to avoid confusion between different usages of superscripts, in this proof we will change $\gamma_r^i$ to $\gamma_i$, i.e. the $i^{th}$ component of the vector $\gamma$, not to be confused with $\gamma_r$, a vector corresponding to the embedding of the $r^{th}$ reasoning mode. Then,

$$\mathbb{E}_{\gamma_i \sim N(0,1)}||\gamma||_2 \cdot ||\beta||_2^p \tag{29}$$

$$=||\beta||_2^p \mathbb{E}_{\gamma_i \sim N(0,1)}\left(\sum_i \gamma_i^2\right)^{\frac{1}{2}} \tag{30}$$

$$\tag{31}$$

We now perform a change of variables and introduce a new random variable

$$z = \sum_i \gamma_i^2. \tag{32}$$

Since each individual $\gamma_i$ is distributed normal, $z$ is distributed according to a $\chi^2$ distribution with $m$ degrees of freedom, and we get

$$\mathbb{E}_{\gamma_i \sim N(0,1)}||\gamma||_2 \cdot ||\beta||_2^p \tag{33}$$

$$=||\beta||_2^p \, \mathbb{E}_{z \sim \chi^2}[z^{\frac{1}{2}}] \tag{34}$$

The expectation now corresponds to $\frac{1}{2}^{th}$ centered moment of the $\chi^2$ distribution with $m$ degrees of freedom, whose k$^{th}$ moments are given as

$$\mathbb{E}_{z \sim \chi^2}[z^k] = 2^k \frac{\Gamma(\frac{m}{2} + k)}{\Gamma(\frac{m}{2})} \tag{35}$$

This ends the proof of the equality.

## B   Program decoder

We provide more details on the program decoder architecture. The hidden size is set to 128 (same as in the VL-Transformer). We use GeLU [3] as non linearity, along with layernorm [1].

**Operations**   The maximum number of operations in one program is set to $N_{maxop} = 9$. The total number of operation's labels is $N_{op} = 212$. We use a one layer GRU [2] with hidden size equals to 128, to infer the operation's hidden embedding $\boldsymbol{h}_i$. It is followed by a two layers MLP ($128 \rightarrow 64 \rightarrow N_{op}$, projecting $\boldsymbol{h}_i$ into a one-hot vector $\boldsymbol{o}_i$.

**Arguments**   Affinity scores $\boldsymbol{a}_{ij}^q$ between each operation's hidden embedding $\boldsymbol{h}_i$ and each token embedding $\boldsymbol{q}_j$ (or $\boldsymbol{v}_j$) are computed with a 2-layer feed-forward network ($256 \rightarrow 64 \rightarrow 1$) from concatenated embeddings. The op arguments are predicted from $\boldsymbol{h}_i$ using another one layer GRU with hidden size equals to 128 followed by a nonlinear projection ($128 \rightarrow N_{maxop}$).

**Hyper-parameters**   are set to $\alpha = 1$, $\beta = 1$, $\gamma = 1$ and $\delta = 100$.

## C   Training details

**Architecture:**   Our VQA architecture is a compact version of the VL-Transformer introduced in [13][1]. In particular, it has 9 language only layers, 5 vision only layers, and 5 cross modal layers. The hidden size is set to 128. In total, this compact version has 26M parameters, allowing to reduce computation time and memory overhead.

**Optimizer:**   All models were trained with the Adam optimizer [7], a learning rate of $10^{-4}$ with warm starting and learning rate decay.

**Oracle transfer:**   is performed following [6]. First, the oracle model is trained during at most 40 epochs on the GQA *balanced* training set with ground-truth image representation. Then we continue the training during 10 epochs, with visual features extracted using an object detector. We use a batch size equals to 196 (96 when using VinVL features).

**BERT/LXMERT [13] pretraining**   is performed during 20 epochs with a batch size of 320 (256 when using VinVL features). All pretraining losses are added from the beginning, including the VQA one. Note that LXMERT [13] is originally pre-trained on a corpus gathering images and sentences from MSCOCO [10] and VisualGenome [8]. In this work, we only train on the GQA [4] *unbalanced* set, with VisualGenome images. After pre-trainning, we finetune on the GQA [4] *balanced* set during 4 epochs, with a batch size of 32 and a learning rate equal to $10^{-5}$.

## D   Computing resources

Training and evaluation has been performed on several compute infrastructures, which include an Nvidia DGX-A100 with $8\times$ A100 GPUs and a cluster with P100 and RTX 2080 GPUs. After design and development, the final training and evaluation runs have been performed on Geforce RTX 2080 GPUs. We provide an estimate for the amount of compute in Table 1 — the number of GPUs and approximate execution times for different models and experimental settings (train, validation and test).

---

[1]We use the code publicly available at `https://github.com/airsplay/lxmert`

Table 1: Training and execution time for one run. *Ours* corresponds to oracle transfer plus program prediction. We also provide the approximated amount of runs done during this work (hyper parameters search, abblation, *etc.*)

| Run | Model | #GPUs | # hours | Total number of runs |
|---|---|---|---|---|
| train | Oracle | 1 | 30 | ≈ 5 |
| train+test | ours 36 RCNN | 1 | 9 | ≈ 100 |
| train+test | ours 100 RCNN | 2 | 10 | ≈ 5 |
| train+test | ours VinVL | 2 | 10 | ≈ 5 |
| train+test | ours 36 RCNN + LXMERT pretrain | 2 | 100 | ≈ 20 |
| train+test | ours 36 RCNN + LXMERT finetune | 1 | 4 | ≈ 50 |
| train+test | ours VinVL + LXMERT pretrain | 3 | 180 | 2 |
| train+test | ours VinVL + LXMERT finetune | 1 | 6 | 2 |

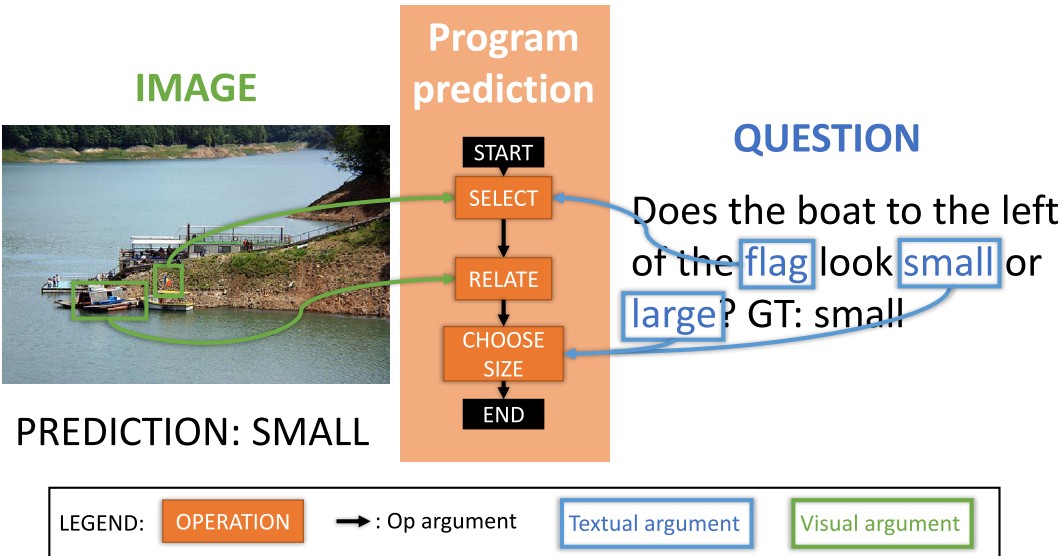

Figure 1: Example of program prediction. The question is: "Does the boat to the left of the flag looks small or large?". Our model (ours+lxmert with VinVL) correctly answers "small".

**C02 Emission** The RTX infrastructure has a carbon efficiency of 0.035 kgCO$_2$eq/kWh. A cumulative of 6500 hours of computation was performed on hardware of type RTX 2080 (TDP of 215W). Total emissions are estimated to be 48.9 kgCO$_2$eq . Estimations were conducted using the `https://mlco2.github.io/impact#compute` presented in [9].

# E  Visualisation of predictions

We provide example of program prediction in Fig. 1 and 2. In Fig. 1, the question is *'does the boat to the left of the flag look small or large?'*. The program decoder successfully infers the correct program. It first predicts the coarse operations – `select`, `relate`, `choose size` –, then adds the arguments taken from the image or the question – boat, flag, small, large –. Finally, the VQA model predicts the correct answer *'small'*. In Fig. 2, the question is *'who is wearing goggles?'*. Similarly to the first example, the program decoder generates coarse operations – `select`, `relate`, `query name` – and visual/textual arguments – woman, who, goggles, wearing–. In these two examples, the decoder correctly predicts that the programs are chains of operations (special case of a tree). At contrary, a question like *'are there nuts or vegetables?'* is a not a chain because of the presence of `exist` and `or` operations.

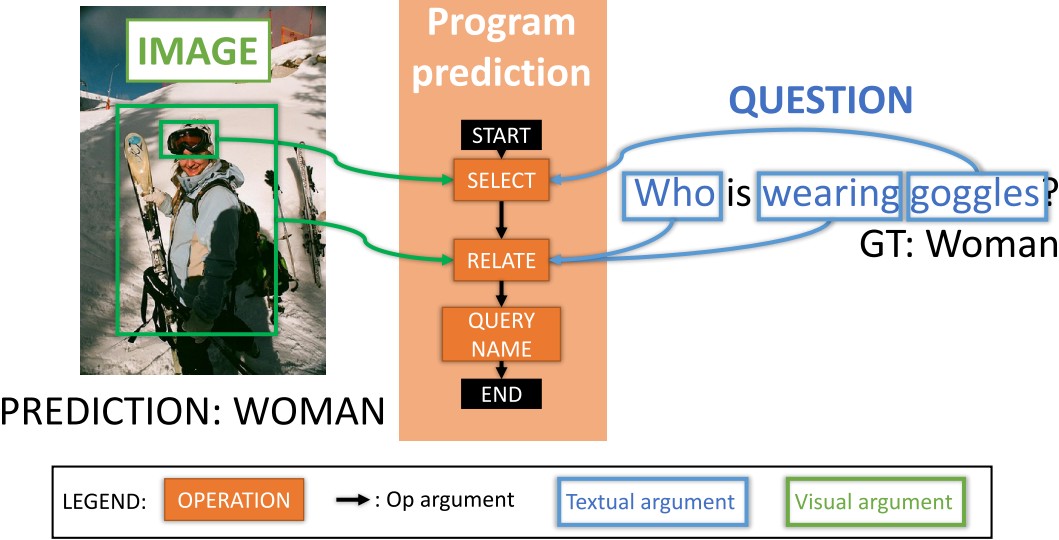

Figure 2: Example of program prediction. The question is: "Who is wearing goggles?". Our model (ours+lxmert with VinVL) correctly answers "woman".

| Ablations | GQA-OOD [5] acc-tail (val.) | GQA [4] val. |
|---|---|---|
| (1) VQA only | 46.9 | 62.2 |
| (2) Random prog. | 45.7 | 61.4 |
| **(3) ours** | **49.9** | **66.2** |

Table 2: Comparison with the *random prog* baseline, where we randomly replace the ground truth program with a program picked from another question (compact model, no LXMERT/BERT pretraining, no Oracle), on GQA val.

## F Sanity check

As a sanity check, and to avoid the unfortunate result pinpointed in [12], we compare our model with a random baseline *random prog.* in Tab. 2. In *random prog.*, we randomly replace the ground truth program with a program picked from another question, during the training. As expected, this random baseline achieves low performances.