# OpenReview forum: "Supervising the Transfer of Reasoning Patterns in VQA"
_NeurIPS.cc/2021/Conference — NeurIPS 2021 Poster_

### Official Review · Reviewer_Ta31 · 2021-07-10

**Rating:** 7
**Confidence:** 4

**Summary:**

The paper studies the problem of training VQA models on noisy/imperfect visual input (e.g., representations from existing object detectors). It proposes using program supervision as an additional task to assist in transferring reasoning patterns learned through clean/oracle visual inputs to the imperfect inputs. The experiments are performed on the GQA/GQA-OOD benchmarks, and the empirical results show gains on both frequent (head) and infrequent (tail) classes, showing improvement in reasoning skills. Furthermore, the work theoretically shows that predicting programs can reduce sample complexity.

**Limitations And Societal Impact:**

The paper acknowledges the limitations of the theoretical work i.e., analysis on MLPs instead of the models actually used in the paper. However, I think there are at least a few other limitations: a) assuming access to oracle visual inputs/programs and, b) lack of empirical studies on sample complexity for pre-training/fine-tuning tasks.

**Main Review:**

Overall, the paper is well written for most parts. The program supervision loss is simple and sensible, but of course requires ground truth programs to be available, which may not always be realistic. The theoretical result relating program supervision to reduction in sample complexity is an important contribution, but the empirical setup could be much better.
I will now present my concerns:

[C1] The phrase: ‘transfer of reasoning patterns’ seems a bit odd for the setup considered in this paper. With such a title/description, I was expecting more focus on the reasoning patterns themselves, instead of visual quality. For instance, the title would be more appropriate for the following scenarios: applying previously learned reasoning patterns to images from new domains or questions phrased differently or to novel concepts/compositions of reasoning steps, but these are not what is being studied here.

[C2] The theoretical result about program supervision reducing sample complexity in reasoning is an important contribution. However, the empirical setup leaves some open questions. Firstly, it would be nice to validate it empirically by fine-tuning on fewer samples. Secondly, does this apply to pre-training stages too? Would it apply to both visual oracle pre-training and BERT-style pre-training? It would be a plus to have those answers or at least have those discussions somewhere.

[C3] Obviously, there are some concerns about the practicality of the proposed method. Firstly, having access to oracle visual annotations is not realistic, yet most of the gains seem to come from them. Second, having access to the ground truth programs is also not realistic. Both of these concerns could be addressed if the sample complexity could be validated empirically i.e., would having program supervision help even if not all instances had oracle visual inputs? Furthermore, how much benefit would it show if only a subset of the instances had program supervision? The results on head/tail do not quite cut it since they were based answer type/answer frequencies if I recall correctly, and may not provide clear analyses based on training phase/type of annotation. I understand that these may require extra experiments, but even some discussions regarding these would help.

[C4] It is important to clarify early on that the program supervision refers to predicting the reasoning operations/arguments, but it is not necessary that the model actually follows that reasoning pattern (or does it?). It would be a big plus if the paper ran MMN [C2], which actually performs those reasoning operations with oracle pre-training too. For instance, if program supervision alone is sufficient and we do not actually need a separate modular architecture to run those reasoning operations, then that would be a very important finding for the community. But this is missing from the paper.

[C5] Some of the details were not completely clear:
I was confused about what ‘reasoning patterns’ meant in the context of this paper, and had to refer to [1] for clarification.
How much does the model trained after oracle pre-training alone achieve?

[C6] Missing citation: HAN [3] could be cited since it is also graph-based and outperforms the other graph-based method mentioned in the paper: NSM [4].

References:

[1] Kervadec, Corentin, et al. "How Transferable are Reasoning Patterns in VQA?." Proceedings of the IEEE/CVF Conference on Computer Vision and Pattern Recognition. 2021.

[2] Chen, Wenhu, et al. "Meta module network for compositional visual reasoning." Proceedings of the IEEE/CVF Winter Conference on Applications of Computer Vision. 2021.

[3] Kim, Eun-Sol, et al. "Hypergraph attention networks for multimodal learning." Proceedings of the IEEE/CVF Conference on Computer Vision and Pattern Recognition. 2020.

[4] Hudson, Drew A., and Christopher D. Manning. "Learning by abstraction: The neural state machine." arXiv preprint arXiv:1907.03950 (2019).

[5] Selvaraju, Ramprasaath R., et al. "Taking a hint: Leveraging explanations to make vision and language models more grounded." Proceedings of the IEEE/CVF International Conference on Computer Vision. 2019.

**Time Spent Reviewing:**

5

---

> ### Author Response · Authors · 2021-08-09
> **Response to reviewer Ta31**
>
> Thank you for your detailed review, with valuable comments and appreciations. Please find the answers of your questions as follows:
>
> 1.**Transfer of reasoning patterns**: The ‘transfer of reasoning patterns’ was introduced in reference [c] below (reference [21] in the paper), it refers to the transfer of the knowledge acquired by the VQA model when trained on favourable conditions (e.g. using oracle visual inputs) to less favourable conditions (e.g. using visual object detected by a pre-trained object detector). It turns out that the visual uncertainty is a bottleneck for this kind of reasoning transfer, motivating our study.
> We argue that this term (“reasoning patterns”) is also particularly adapted to our paper, as we motivate the decreased sample complexity through program supervision by the decomposition of the global reasoning function into different reasoning modes, which can intuitively be linked to “patterns of reasoning” dependent on the question. But in fine, we think this is a term only and as such is debatable, of course.
> However, the idea of analysing the transfer of reasoning patterns in the context of domain adaptation is very interesting, and we will discuss it in the conclusion.
>
> 2.**Experimental evidence of the decreased sample complexity**: We performed these experiments for the rebuttal, see our general answer to all 4 reviewers. We will add more discussion about it in the paper.
>
> 3.**Extra-annotation**: The oracle visual annotation is exactly the same as the one used for the supervision of the object detector itself. It consists in the ground truth objects present in the image, cf. reference [c] below (reference [21] in the paper). Therefore, we can assume that having access to such annotation is realistic, especially because SOTA models already use this annotation (as they rely on a pre-trained object detector). Nevertheless, we agree that having access to ground truth programs is less realistic, as it is more costly to annotate (as stated in L337). The goal of our study was to provide empirical and theoretical evidences that the program supervision can benefit to the VQA performances. Next step will be to adapt our method to settings where such annotation is rare or incomplete. As an illustration, we already provide an experiment showing that the program supervision allows to outperform the baseline while using much less training data (see our general answer to all 4 reviewers). In addition, we will add more discussion about this in the conclusion.
>
> 4.**Difference with MMN**: Indeed, there is a strong difference on how the program prediction is used in our method vs. in MMN, and we will clarify it in the paper. In our method, the purpose of program supervision is to encourage the VL-Transformer to encode the reasoning information into its (textual and visual) hidden embeddings, in order to improve its reasoning ability. But it is a ‘soft’ constraint, as the VQA prediction does not necessary have to follow the predicted program. At contrary, MMN uses the program supervision to learn translating the question into a program, which is then applied to the visual objects. Therefore, the MMN hidden embeddings, from which the answer is predicted, does not have to encode the reasoning information. In this paper, we specifically demonstrate (empirically and theoretically) the benefit of encoding the reasoning information into the embeddings layers. We agree that comparing the benefits of these two different types of program supervision is an interesting (and important) line of work. We will add it in the discussion and propose it as future work.
>
> 5.**Oracle details**: After oracle pre-training alone, on the validation set, the model reaches 93.4% when fed with oracle data and 58.8% (cf. reference [c] below (reference [21] in the paper). when fed with Faster RCNN visual objects. As comparison, on the validation split, oracle pretraining plus finetuning on Faster RCNN objects reaches 66.3% (corresponds to score (b) in our Table 1), whereas oracle pretraining plus finetuning plus program supervision reaches 67.4% (corresponds to score (c) in our Table 1). We can clearly see the benefit added by the program supervision.
>
> 6.**Does this [decreased sample complexity] apply to pre-training stages too? Would it apply to both visual oracle pre-training and BERT-style pre-training?**: For BERT-style pre-training (actually, LXMERT style pre-training,  which is the BERT like variant for VQA) this has been shown in the paper in table 1 (section “+lxmert”). In principle it should also apply to Oracle pre-training only, but testing it was beyond the scope of the paper.
>
> 7.**Missing references**: thank you for providing these references, we will add them in the camera ready version.
>
> [c] Kervadec, Corentin, et al. "How Transferable are Reasoning Patterns in VQA?." Proceedings of the IEEE/CVF Conference on Computer Vision and Pattern Recognition. 2021.

---

### Official Review · Reviewer_uBf9 · 2021-07-16

**Rating:** 7
**Confidence:** 3

**Summary:**

This paper describes a novel approach for doing more robust VQA that doesn’t just learn to exploit the biases in the data. They build on previous work which demonstrated that training a model on oracle visual data makes it less susceptible to learning biases in the data. While the previous work showed that the better reasoning skills learnt by the models trained on oracle visuals decline when transferred to the noisier visuals extracted by pre-trained vision models, here the authors demonstrate that adding the additional supervision step of making the model predict the reasoning program alongside the VQA answer makes it more robust to such distributional shifts.


**Limitations And Societal Impact:**

Yes

**Main Review:**

The paper is exceptionally well written. It provides both theoretical and empirical results, and discusses ethical implications and risks well.

The only problem is that it doesn’t achieve SOTA VQA results. Yes, the SOTA model (OSCAR [39]) requires 50 times more images, but the proposed model requires 2x more sentences and the additional program supervision, which is hard to obtain as the authors acknowledge. Saying this, it does better than the comparable baseline with program supervision trained on the same amount of data.

The authors show theoretical results suggesting that their additional program supervision results in better sample complexity by learning re-usable reasoning modules implicitly. This is probably the most interesting result of the paper, and I would have liked to see it explored more experimentally. For example, I am not sure what experimental results support the sample complexity claim. Also, could the authors maybe use the program module to analyse the failure cases? Does it bring extra interpretability to the model?

The program prediction module is entirely independent from the VQA module. Why do the authors believe that it helps regularise VQA learning? Have they considered making VQA answer module dependent on the reasoning performed in the program module?


**Time Spent Reviewing:**

2

---

> ### Author Response · Authors · 2021-08-09
> **Response to reviewer uBf9**
>
> Thank you for your detailed review, with valuable comments and positive feedbacks (e.g. “the paper is exceptionally well written”). Please find the answers of your questions as follows:
>
> 1.**OSCAR uses 50x more images**: It is true that our method requires 2x more sentences. However, these sentences are automatically generated, that is why the cost is limited. On the other hand, OSCAR requires 50 times more real images with annotations, implying a higher cost. Beside, as also said in the paper (L337), we agree that program annotation can be costly to collect. Therefore, we would like to insist on the fact that the goal of our study was to show how program supervision can help to transfer the reasoning knowledge acquired in favourable conditions (oracle). The next step will be to adapt our method to settings where the program annotation is rare or incomplete while being competitive with SOTA, but we leave it as future work.
>
> 2.**Experimental evidence of the decreased sample complexity**: We performed these experiments for the rebuttal, see our general answer to all 4 reviewers.
>
> 3.**Using the program module to analyse the failure cases**: We did not conduct such extensive analyses of failures cases. However, we have observed that many failures are caused by visual objects not (or falsely) detected by the detector. In these cases, the predicted operations are correct, but the module fails to associate the correct visual arguments.
>
> 4.**Why does program supervision regularise VQA learning?**: Under the hypothesis that spurious biases exist in the training data, which can (and often will) lead to learning wrong shortcuts, the additional supervision of reasoning programs can help the model to learn the correct reasoning steps. Yes, we agree that the two heads are independent, which is the case of most (all?) auxiliary losses, be it for VQA or other tasks, for instance BERT / LXMERT like losses. In our case, program supervision enforces the VL-Transformer to encode information about reasoning into the textual and visual embeddings. While there is no guarantee that this information is used by the VQA part, we argue that the functional complexity of the reasoning program has been reduced, making shortcuts less “necessary”. This has been formalized in the theoretical part of the paper, in particular Assumption 3 and the explanations given in L243-247. We will try to reexplain it here in different words:
> For each instance (question + image pair) the additional supervision enforces the encoding of the reasoning program in the hidden embeddings of the VL-transformer. We would like to insist that this means the particular (individual) reasoning program for the given sample at hand. Consequently, later layers of the VL-transformer can, in principle, establish a direct correspondence between this latent representation of the reasoning program and the function learned by these layers, which means that by gradient descent we learn parameters of the later layers which “only” express the information already provided in the latent layers. We have shown in the theoretical part, that learning these single reasoning modes individually is easier than learning the full reasoning program, where a much more complex program needs to be learned from a more complex input, namely the question + image pair directly. As summary, while shortcut learning has not been made impossible, the difference in functional complexity between short cuts and required reasoning has been substantially reduced.
>
> 5.**Making VQA answer module dependent on the reasoning performed in the program module**: This is an interesting line of work, which have also been proposed by reviewer Ta31. We will discuss it in the conclusion and propose it as future work.
>
> [b] Chen, Wenhu, et al. "Meta module network for compositional visual reasoning." Proceedings of the IEEE/CVF Winter Conference on Applications of Computer Vision. 2021.

---

> > ### Comment · Reviewer_uBf9 · 2021-08-19
> > **Score unchanged**
> >
> > Thank you for your detailed response. Since my score was quite high to begin with, I am leaving it unchanged.

---

### Official Review · Reviewer_NU5A · 2021-07-17

**Rating:** 7
**Confidence:** 3

**Summary:**

This paper proposes a new loss terms to encourage Transformer-based models perform "explicit reasoning", which is beneficial for transferring knowledge from oracle-trained VQA models to deployable ones. Specifically, the novel loss is similar with the one commonly used in modular network (e.g. converting questions to executable program/functions), but instead of relying on the predicted programs to assemble executable programs during inference, in this paper this loss is only used during training as an auxiliary regularization on top of regular Transformer-based VQA models. The authors provide theoretical proof about the benefits of using the additional loss, and empirically evaluated the advantage on GQA dataset (including out-of-distribution evaluation on GQA-OOD). The proposed system demonstrates not the best but still competitive performance especially considering the model size and training efficiency (without using extra data).

========================================================================================

Thanks for the response from the authors. I've updated my rating after reading the rebuttal as well as the comments from other reviewers.

**Limitations And Societal Impact:**

- I didn't find obvious mistakes from the theoretical analysis (I'm not an expert though), but it seems from the experimental validations there is no observation to directly or indirectly (with a measurable proxy) support the claim? More specifically, is there any metrics can be used to provide more straightforward evidence about the statement "... such program prediction can lead to decreased sample complexity ..." (from the abstract).

- One probably missing ablation about the proposed loss term is to directly train a VQA system with the loss from scratch but without oracle transfer. I feel this can be a good evidence to verify whether the proposed regularization is only helpful for knowledge transfer or generally can push the Transformer models to better handle the reasoning.

- Some minor issues: (1) why in Table 1 the accuracy numbers of binary, open and test-std for baseline models are missing? (2) L294 mentions an accuracy of 58.8 on testdev, when program supervision+lxmert is used but not oracle transfer. The oracle transfer+lxmert seems to be 58.4 according to the table, which is against the claim of L295 ("this is lower than oracle transfer's accuracy...")?

**Main Review:**

1. Originality
- This work is an extension following a very recent work (paper [21] in the reference) and the goal is to improve the knowledge transfer effectiveness from oracle-trained VQA systems to deployable (e.g. visual signals coming from upstream visual recognition systems), rather than directly training VQA models. This is a promising direction for building future VQA models.
- The proposed loss term is similar to existing work from modular networks (e.g. paper [6]), but the usage is different: in previous work the predicted executable program is used for producing execution graph especially during inference, but in this work the entire module is only used as regularization during training.
- The authors provide theoretically analysis about their novel regularization term on a simplified framework (MLP).

2. Quality
- I don't see any obvious technical flaws in this paper, and the experiments are quite extensive with proper ablations.
- One typo in L154: I assume the loss should be $L_{dep}$ rather than $L_{varg}$ (which is already defined in L149).

3. Clarity
- This paper is well-written and easy to follow. The main idea is straightforward, and the experimental sections are clear for readers to reproduce.
- I only quickly checked the math but not carefully verifying all the details in the proof (I'm not an expert in theory).

4. Significance
- This paper explores a novel approach (following a recent work) of building VQA systems, which aims at leveraging oracle-trained VQA systems (with perfect visual signals) and transferring the knowledge to accommodate noisy visual inputs. The high-level idea is different from the tradition (training from scratch with noisy visual signals) and the promising results may attract more future research efforts.

**Time Spent Reviewing:**

10

---

> ### Author Response · Authors · 2021-08-09
> **Response to reviewer NU5A**
>
> Thank you for your detailed review, with valuable comments and positive feedbacks (e.g. “this is a promising direction for building future VQA models”). Please find the answers of your questions as follows:
>
> 1.**Experimental evidence of the decreased sample complexity**: We performed these experiments, see our general answer to all 4 reviewers.
>
> 2.**Program supervision w/o oracle**: We actually did this experiment (L295) and obtained an accuracy of 58.8%. This is lower than the accuracy obtained when adding the oracle transfer to the program supervision (59.3%, cf. Table 1). This suggests that the program prediction is helpful during the transfer. We will reformulate the sentence L295 to make it more explicit.
>
> 3.**Test-std score for the baseline**: GQA test-std is a hidden test set, which is restricted to a limited number (10) of submissions (https://eval.ai/web/challenges/challenge-page/225/submission). Given this constraint, we only provided the test-dev accuracy for the baseline (in Table 1), as it is outperformed by a large margin by our method.
>
> 4.**L294**: Our sentence is not clear, we apologize and we will reformulate as “This is lower than oracle transfer *plus program supervision* accuracy, demonstrating the complementary of the two methods.”
>
> 5.**Typo L154**: Thanks for notifying us, we will correct it.

---

### Official Review · Reviewer_8H1F · 2021-08-01

**Rating:** 5
**Confidence:** 4

**Summary:**

This paper targets on the application problem of visual question answering, whose recent state of the art was set by attention (typically transformer) - based deep neural networks. This work propose to use program supervision to introduce extra loss terms in the training of these transformer based models, arguing that the guidance from these signals improve the model's robustness against noise and fluctuations in the visual input during transfer learning. Theoretical analysis and experimental evaluations are conducted on program supervision, leading to plausible findings and results.

**Limitations And Societal Impact:**

Briefly discussed.

**Main Review:**

This paper discusses a fundamental problem of how to assess and improve the "reasoning" capabilities of neural networks. In general, this problem is challenging to tackle as reasoning and pattern recognition are strongly entangled in most end-to-end networks with multi-modal inputs. On this note, using program supervision to guide representation learning and then perform transfer learning on more noisy inputs seems a plausible direction.

My questions challenges to this paper are listed as follows:

1. It seems to me that the CLEVR dataset is specifically designed for addressing a very similar if not the same issue of neural network being dominated by perceptual bias in VQA. Therefore it is the most natural dataset to study under the context of this paper. Is there a good reason why this dataset is not used?

2. This paper claims that program supervision improves the robustness against VISUAL inputs and surround all discussions and experiments around this claim. While the results seems plausible, it remains skeptical to me why this is the case. In most cases, programs are mainly depend on the question, not the visual context (in CLEVR and GQA, the program is one-to-one mapped to the question). Hence the program supervision will mostly lead to a better question parser rather than a more robust visual branch.

3. Following 1 and 2, I believe it would be more natural to study transfer learning against noise and biases in the language space rather than the visual space.

My overall score to this paper is "Marginally below threshold". No further experiments are requested in rebuttal.

**Time Spent Reviewing:**

3

---

> ### Author Response · Authors · 2021-08-09
> **Response to reviewer 8H1F**
>
> Thank you for your detailed review, with valuable comments and appreciations. Please find the answers of your questions below:
>
> 1.**CLEVR dataset**: While we do agree that in principle CLEVR is well suited for evaluating the reasoning ability independently of perceptual biases, we do think and claim that GQA is even more suited for this kind of work. As a purely synthetic dataset, CLEVR provides a limited diversity only on the vision side, which make it less challenging, and it starts to saturate. SOTA models already reach an accuracy above 99% (ref [a] below). On the other hand, GQA also focuses on evaluating reasoning with similar regularities in questions and object relationships, but it is based on real images and manually annotated scene graphs. This allows evaluating the reasoning ability in a more realistic environment. In particular, GQA puts more strain on the models’ generalization capabilities than CLEVR. As a result, SOTA models only reach 65% on GQA.
>
> 2.**Robustness against visual input only**: this might be a misunderstanding. While the question fully determines the reasoning mode, i.e. the logical operations necessary to answer the question, the supervision of the program is even stronger, as the supervised reasoning program is more fine grained and includes some details on vision. We will here briefly recall the definition of a reasoning program: program supervision includes the prediction of operations (“choose color”, “verify size”, “relate”, etc…) and their arguments. Arguments can be question words or visual objects. As a result, the programs depend on both vision and language. In Table 2, we show that adding the supervision of visual arguments provides the highest gain in accuracy (compare row 5 vs. 4). Thus, we conclude that the program supervision improves the robustness against visual input.
> We will include a discussion of this point in the paper.
>
> 3.**Robustness against textual input**: Studying robustness against language variations is indeed a promising direction or research. However, in this paper, we study the reasoning ability of the VQA model by evaluating its accuracy in OOD settings (using GQA-OOD). These OOD settings take into account the context of the question-image pair and the rareness of the answer given that context. It is then dependant on both vision and language variations. As a consequence, even if we do not address the specific issue of linguistic robustness, our OOD evaluation includes both visual and language robustness.
>
>  [a] Yi, Kexin, et al. "Neural-Symbolic VQA: Disentangling Reasoning from Vision and Language Understanding." NeurIPS. 2018.

---

### Author Response · Authors · 2021-08-09
**General answer to all 4 reviewers**

We thank the reviewers for their efforts and their reviews of high quality. We are happy that they appreciated:
- Our valuable contribution: “*this paper discusses a fundamental problem of how to assess and improve the reasoning capabilities of neural networks*”, “*provides both theoretical and empirical results*”, “*the theoretical result relating program supervision to reduction in sample complexity is an important contribution*“;
- The novelty of findings: “*The high-level idea is different from the tradition*”, “*the promising results may attract more future research efforts*”;
- The quality of writing: “*well-written and easy to follow*”, “*the paper is exceptionally well written*”, “*discusses ethical implications and risks well*”;
- The quality of experiments: “*the experiments are quite extensive with proper ablations*”, “*the experimental sections are clear for readers to reproduce*”, “*I don't see any obvious technical flaws in this paper*”;

**We answered each reviewer separately using the open-review discussion tool. However, we will answer one suggestion, which has been raised by several reviewers, namely the request for a more direct empirical evaluation of the theoretical claim on sample complexity.**

Our theoretical findings demonstrate that the program supervision reduces the sample complexity of VQA training. The fact that adding program supervision on top of the VQA baseline improves its performance in in- and out-of-distribution settings (cf. Table 1 in the original paper) is a first empirical evidence supporting the theoretical claims.

However, we agree that a more direct evaluation might be of interest. Therefore, we propose an additional experiment, where we vary the amount of training data from 20% to 100%, comparing overall accuracy obtained with and without program supervision. As requested, in this setup, we do not use oracle transfer neither LXMERT pretraining. Results are in the Table below. We observe that adding program supervision allows to reach an accuracy similar to the baseline while using less data. For instance the program supervision allows to outperform the baseline (trained on 100% of the data) while using only 40% of the training data. This is an additional empirical evidence supporting the theoretical claim, we will add it in the paper.

|                      | 100% | 60%  | 40%  | 20%  |
|----------------------|------|------|------|------|
| baseline             | 57.3 | 53.2 | 51.5 | 48.8 |
| base. + program sup. | 65.9 | 63.0 | 60.5 | 53.6 |

---

> ### Comment · Reviewer_NU5A · 2021-08-17
> **Why the new experiments are "... more direct empirical evaluation of the theoretical claim on sample complexity"?**
>
> Thanks for the additional experiments, although I was not sure why this "varying amount of data" experiments can serve as a "more direct evaluation". It seems to me that these experiments make sense but are not directly supporting "decreased sample complexity" claim. Any insights/explanations why a more direct evaluation (empirically measuring sample complexity, rather than showing the overall accuracy) is infeasible?

---

> > ### Author Response · Authors · 2021-08-19
> > **Experimental evidence of the decreased sample complexity**
> >
> > We are not sure which experiments could give better empirical evidence for lower sample efficiency, but since reviewer NU5A explicitly mentioned our usage of accuracy as a measure for sample efficiency, we will try to explain our point of view differently, by transposing the table describing our experiments. We agree that the way we phrased the explanations might have not been direct enough.
> >
> > Sample complexity characterizes the minimum amount (=$M$) of samples necessary to learn a function with sufficiently low $(=\epsilon)$ error with sufficiently high $(=\delta)$ probability. If we fix an empirical measure of success of learning as a certain threshold on accuracy, then we can measure sample efficiency as the amount of samples (as a proportion of the full training set) necessary to reach this accuracy. This is exactly what we did in the experiments provided in the rebuttal, but, agreed, we had presented it in slightly different way.
> >
> > In the “varying amount of data” experience, we decrease the amount of training data (=$M$) and measure the resulting accuracy (related to $\epsilon$). We observe that for a given target accuracy (e.g. >55%), the number of required training samples (=$M$) is lower when using our program supervision method (30% vs. 100% of the data). Therefore, we think that this supports the “decreased sample complexity” claim. As a side note, this experiment is similar to the one performed in Figure 4 of [35] (referenced in the paper). Does NU5A agree with this explanation?
> >
> > We rearranged the table to make the message clearer. The first column corresponds to the target accuracy $(=\epsilon)$. Then, we measure what is the amount of training data (=$M$) required to reach the target for each baseline (2nd and 3rd columns). *Baseline + program* supervision requires less training data:
> >
> > | Target accuracy |        baseline       |  base. + program sup. |
> > |:---------------:|:---------------------:|:---------------------:|
> > |     acc.>65%    |     not reachable     | 100% of training data |
> > |     acc.>60%    |     not reachable     |  40% of training data |
> > |     acc.>55%    | 100% of training data |  30% of training data |
> > |     acc.>50%    |  40% of training data |  20% of training data |
> > |     acc.>45%    |  10% of training data |  20% of training data |
> >
> > [35] Xu, Keylu, et al. "What Can Neural Networks Reason About?." ICLR 2020 (2020).

---

> > > ### Comment · Reviewer_NU5A · 2021-08-25
> > > **Thanks!**
> > >
> > > Thanks for the extra explanations! Now I think I better understand this concept.
> > > If I'm not mistaken, here the "sample complexity" is similar with "dataset diversity", i.e., it does not measure the "complexity" of individual sample, but more like the "diversity" as a characteristic of the dataset as a whole?

---

> > > > ### Author Response · Authors · 2021-08-26
> > > > **Sample complexity**
> > > >
> > > > Thank you for this discussion and your continuing interest in our work.
> > > >
> > > > Yes, you are right that the concept of “sample complexity” could roughly be linked to dataset diversity, in the sense that more diverse data requires more samples for learning. We think that an even better word would be the “difficulty” of the decision function. Intuitively, and through an example, if the labels Y are linearly dependent on the input X, then this would be quite easy to learn compared to complex non-linear dependencies, and it would require less samples.
> > > >
> > > > More formally, we use the concept “sample complexity“ introduced by Valiant et al. [a] for PAC learning, and whose definition we had reproduced in Appendix A (supplementary material, definition A.1). In this context, sample complexity depends on several components: the learning algorithm itself, the neural model (i.e. the hypothesis class / network architecture in the case of deep networks) and the complexity of the decision function. Classical theory often links sample complexity to the capacity of the model, e.g. it’s VC-dimension [b][c], which is (unfortunately) independent of the data.
> > > >
> > > > To cope with this, we use the modern estimators of sample complexity developed for the deep learning era (see [d] for an overview), which provide the possibility of calculating tighter bounds under the assumption that learning is performed by over-parametrized deep networks and stochastic gradient descent. These estimators are data-dependent and as such more powerful.
> > > >
> > > > Within this framework, in particular the work of Arora et al (reference [26] in the submitted paper / [e] below), sample complexity is linked to the functional form of the decision function directly. If the functional form is simpler, learning it requires less samples. [26] provides a direct way to estimate sample complexity, if the functional form is known, or through its estimation from training data in the form of a stochastic Gram matrix.
> > > >
> > > > In our paper we show, under some assumptions, that program prediction allows to learn a set of individual functions independently, each of which is simpler (in the sense of sample complexity), compared to learning the full function (the baseline setting).
> > > >
> > > >
> > > >
> > > > [a] L.G. Valiant. A theory of the learnable. In Communications of the ACM, volume 27(11), 1984.
> > > >
> > > > [b] V.N. Vapnik   and A.Y. Chervonenkis, On the uniform convergence of relative
> > > > frequencies of events to their probabilities, Theory of Probability and its applications, 1971.
> > > >
> > > > [c] Shai Shalev-Shwartz and Shai Ben-David, Understanding Machine Learning: From Theory to Algorithms, Cambridge Press, 2014
> > > >
> > > > [d] M. Belkin, Fit without fear: remarkable mathematical phenomena of deep learning through the prism of interpolation, Acta Numerica, VoL. 30, Cambridge University Press, 2021.
> > > >
> > > > [e] S.S. Du S. Arora, W. Hu, Z. Li, and R. Wang. Fine-grained Analysis of optimization and generalization for overparametrized two-layer neural networks. ICML, 2019.

---

> > > > > ### Comment · Reviewer_NU5A · 2021-08-27
> > > > > **Thanks for the follow-up discussions!**
> > > > >
> > > > > Thanks for the follow-up discussions, now I have a better understanding about the situations here. I would like to keep my initial, positive rating. Good luck!

---

### Decision · Program_Chairs · 2021-09-27

**Decision:**

Accept (Poster)

**Comment:**

The authors addressed most of the reviewers concerns and all reviewers who engaged in post-rebuttal discussion recommend to accept the paper, one reviewer increasing their score.
The paper contributes an interesting approach to transfer reasoning patterns with program supervision in the context of visual question answering and a solid evaluation.

I recommend accepting the paper under the expectation that the authors will address the concerns as promised in the author response, including:
1) The empirical evaluation of the theoretical claim on sample complexity
2) improve clarity, typos
3) discuss related work Chen et al.

If possible, please also include a better failure analysis as suggested by reviewer uBf9.